# The Untrained Response of Pet Dogs to Human Epileptic Seizures

**DOI:** 10.3390/ani11082267

**Published:** 2021-07-31

**Authors:** Neil A. Powell, Alastair Ruffell, Gareth Arnott

**Affiliations:** 1School of Biological Sciences, Queens University Belfast, Belfast BT9 5DL, UK; g.arnott@qub.ac.uk; 2Search and Rescue Dog Association Ireland North, Newcastle BT33 0PW, UK; 3School of Natural and Built Environment, Queens University Belfast, Belfast BT7 1NN, UK; a.ruffell@qub.ac.uk

**Keywords:** epilepsy, untrained seizure alert dogs, seizures, olfaction, volatile organic compounds, remote odour delivery mechanism, trigger mechanism, early warning

## Abstract

**Simple Summary:**

Anecdotal accounts abound of pet dogs predicting their owner’s epileptic seizures by becoming attentive and by demonstrating attention-seeking behaviours, but no scientific study has investigated the veracity of these claims. Here, we explored this phenomenon, by assuming the presence of seizure-associated odours and then recording the reactions of a cohort of pet dogs to the emergence of such odours, apparently coming from their non-epileptic owners. Using two specially designed pieces of apparatus called the Remote Odour Delivery Mechanism (RODM), we separately delivered epileptic seizure-associated odours and nonseizure associated odours and video-recorded the reactions of the dogs to each. We found that all the dogs demonstrated more affiliative behavioural changes when confronted by seizure-associated odours, compared with their response to control odours. Our results support the view that untrained dogs detect a seizure-associated odour and are in line with the findings of the emerging literature, which attests that those epileptic seizures are associated with a unique volatile organic signature.

**Abstract:**

Epilepsy is a debilitating and potentially life-threatening neurological condition which affects approximately 65 million people worldwide. There is currently no reliable and simple early warning seizure-onset device available, which means many people with unstable epilepsy live in fear of injury or sudden death and the negative impact of social stigmatization. If anecdotal claims that untrained dogs anticipate seizures are found to be true, they could offer a simple and readily available early warning system. We hypothesized that, given the extraordinary olfactory ability of dogs, a volatile organic compound exhaled by the dog’s epileptic owner may constitute an early warning trigger mechanism to which make dogs react by owner-directed affiliative responses in the pre-seizure period. Using 19 pet dogs with no experience of epilepsy, we exposed them to odours that were deemed to be characteristic of three seizure phases, by using sweat harvested from people with epilepsy. The odours were delivered to a point immediately under a non-epileptic and seated pet dog owner’s thighs. By altering the alternating odours emerging from sweat samples, captured before seizure, during a seizure and after a seizure, and two nonseizure controls, we were able to record the response of the 19 pet dogs. Our findings suggest that seizures are associated with an odour and that dogs detect this odour and demonstrate a marked increase in affiliative behaviour directed at their owners. A characteristic response of all 19 dogs to seizure odour presentation was an intense stare which was statistically significant, (*p* < 0.0029), across the pre-seizure, seizure and post-seizure phases when compared to control odours of nonseizure origin.

## 1. Introduction

International surveys investigating whether untrained pet dogs can predict the onset of seizures in humans with epilepsy, have reported that they demonstrate behavioural changes which resemble attention seeking. The behaviours include intense staring; maintaining close-proximity; excessive panting; paw lifting; vocalisation; increased locomotor activity; licking; yawning and scratching [1,2,3,4,5,6,7,8]. While some authors contend behaviours of that nature might be indicators of physiological canine stress [9], others, such as [10,11,12], conjecture that stress in dogs is more likely to be demonstrated in activities such as cringing; crouching; freezing; hiding; shaking or barking; growling; baring teeth; snapping and lunging [13]. In the [8] international survey of untrained reactions of dogs to seizures, affiliative behaviours were the most commonly reported, with fearful or aggressive behaviours observed in less than 2% of dogs.

Here we investigate the hypothesis that seizures are associated with distinct Volatile Organic Compounds (VOCs) and that these are the trigger mechanism for behavioural changes seen in untrained pet dogs. The proposition for the existence of distinct seizure-associated VOCs is supported by reports of physiological changes and excessive electrical activity in the brain preceding epileptic seizures [14,15,16]. Previous research indicates that these are instigated by the autonomic nervous system and the hypothalamic-pituitary-adrenal axis (HPA) and cause an increase in heart and respiration rates [17,18,19,20]. VOCs, which dissolve in the blood and saliva are exhaled as part of the respiratory process, or effused as sweat emanations [7,21,22,23,24,25,26,27,28,29]. VOCs have previously been shown to be indicators of a range of diseases including the presence of cancers, cholera, cystic fibrosis, diabetes, gut diseases, heart allograft rejection, heart disease, liver diseases, pre-eclampsia, renal disease, TB and congestive heart failure in older patients and COPD [24,26,30,31,32,33].

Given the extraordinary scent capabilities of dogs [34,35,36,37,38,39], it is logical to hypothesize that VOCs may also act as a trigger mechanism for alterations in the behaviours of untrained pet dogs at the onset of epileptic seizure. Dogs are known to communicate not just between themselves but also with humans, using strategies which include visual, tactile, auditory, and olfactory signals [4,40]. They are known to gaze intensely when trying to draw the attention of their human care provider to a difficult to reach object [41,42,43,44], and can understand that a pointed finger indicates a direction they should follow [45,46]. Therefore, if the many anecdotal reports of untrained pet dogs anticipating seizures and apparently attempting to communicate that to their caregiver, are true, a seizure-specific VOC stimulus for such behavioural change seems feasible.

The emergence of three recently published studies provides compelling evidence to support this hypothesis; for example, [7] report that medical detection dogs were successfully trained to alert on seizure-specific odour(s) and later [28] attest that data retrieved from SIFT-MS chemometric analysis illustrated the presence of pre-seizure-associated VOCs. Similarly, [29], delineate that seizure detection dogs demonstrated a predictive ability for seizures of up to 90 min prior to a seizure event.

The aim of this current study was to explore, at a fundamental level, whether dogs with no previous training for epilepsy detection and having never previously witnessed an epileptic seizure would show changes in behaviour when exposed to odours associated with human epileptic seizures and apparently emerging from their owners.

## 2. Methods

A repeated measures design experiment was conducted in which 19 recruited non-epilepsy dog-owner dyads from a local dog training club were subjected to a series of odours from sweat samples from three volunteer people with epilepsy and sweat samples from two non-epileptic (control) volunteers. Because this experiment included working with human volunteers, all methods were performed in accordance with the relevant guidelines and regulations.

### 2.1. Ethical Note

Ethical approval for the study was given by the Research Ethics Committee, School of Biological Science Queens University Belfast,
To conduct nonlicensed animal research using 19 pet dogs which were exposed to seizure and control sample odours and their reactions monitored and recorded to those odours, Ref No. QUB-BS-AREC-19-001.Ethical approval was also granted to obtain sweat samples from three epilepsy patient volunteers and three control volunteer participants and for 19 dog owners to accompany their dogs and sit passively while the reaction of their dogs to seizure and control odours were being recorded. REF: 04/19/PowellNR1.

All samples were provided by participants who had completed Informed Consent Forms. The passive observation and recording of the dogs’ responses to seizure and control odours were conducted in accordance with relevant ‘Arrive’ guidelines and regulations, (https://arriveguidelines.org, (accessed on 1 September 2019)).

### 2.2. Volunteers for Sweat Samples

An internet search generated a list of the principal English-speaking epilepsy Charities in the UK and Ireland and was extended internationally to involve volunteers from a broad cultural basis, thereby minimizing the risk of location bias. Appeals were also made for participants via the social media channels of Queen’s University Belfast. Charities were informed that the aim of the study was to scientifically investigate reports of pet dogs apparently anticipating seizure onset in their owners. To facilitate this research project, volunteers had to have medically diagnosed epilepsy, experience frequent seizure events (daily, weekly) and own pet dogs that could predict seizure onset.

This yielded three epilepsy volunteers from different areas of the UK and Ireland, each of whom owned a dog that demonstrated pre-seizure awareness behaviour and remained with its owner throughout their seizure. Warning times varied from 10 min to 60 min. Early warning behaviours by these dogs enabled the acquisition of pre-seizure samples which would otherwise have been difficult to acquire. All three epilepsy volunteers, were females within the age range 21–55:–Volunteer A—a mature female living with daily absence and tonic-clonic seizures. Cause of epilepsy unknown.–Volunteer B—a mature female, also experiencing daily absence and tonic-clonic seizures. Cause of epilepsy unknown.–Volunteer C—an adult female with the genetic Lennox–Gastaut syndrome and severe developmental and learning issues and experienced several daily recurring tonic-clonic seizures. She was cared for by her parents.

Control samples were provided by two people who did not have epilepsy, had no contact with people living with epilepsy, and were dog owners. It was felt that both groups should own dogs to avoid potentially confounding factors regarding a dog/no-dog effect.

The epilepsy and control groups were asked not to alter their normal bathing/showering habits for sampling purposes because it was felt this would more accurately reflect the everyday conditions within which their own dogs reacted to seizures. Both groups were given written instructions on how to capture and store the apocrine sweat samples using sterile gauze pads (see Appendix A and Appendix B). Previous studies exploring the efficacy of bio-medical detection dogs, attempted to capture VOCs in exhaled breath and sweat taken from the back of the neck and hands [7,47]. However, apart from the difficulties in maintaining breath sample integrity, reservations have been expressed about the efficacious nature of sweat samples from hands and neck [27,48]. Thus, this study elected to use sweat samples taken from the axillae as the most appropriate vehicle for the detection of biomarkers [27]. For those with epilepsy, trusted others were invited to assist in harvesting samples during the course of the three seizure phases (one sample for each seizure phase)
–pre-seizure taken when their untrained pet dog characteristically indicated that a seizure was imminent,–seizure sample harvested immediately while a seizure was occurring–post-seizure taken 6 h after a seizure episode to allow time for potential seizure-associated odours to dissipate.

### 2.3. Sample Collection and Storage

All samples were stored at 4 °C since this method has been found to retain VOCs without negative impact for up to 4 weeks [7,26,28,29,30,49,50,51]. To minimize the risk of degradation of VOCs, the airtight glass vials were recapped between each sample presentation and only 3–4 dogs were tested each day [29].

### 2.4. Study Participants

Canine participants and their owners were recruited in part from friends and acquaintances, and from a dog Training Club in Lisburn Co. Antrim (Lisburn, Northern Ireland), following a routine training club night, when time was set aside for the research project to be explained. Those present were invited to participate with their dogs in an investigation which would explore whether epileptic seizures are accompanied by a distinct odour. They were also informed that if evidence could be found for the existence of such an indicator, it could offer profound benefits for the safety of people who have difficulty managing their epilepsy. This appeal yielded a sample of 19 dogs of varying breeds, ages, and of both genders, whose owners were mainly female (16) (male, 3) with an age range across both genders of 18–66+ (Table 1). We hypothesized that, if pet dogs owned by people with epilepsy anticipate seizures by showing owner-directed behavioural changes, then pet dogs owned by non-epileptic people, could also demonstrate an attention-seeking behaviour on encountering seizure-associated odours apparently coming from their owners. In designing this experiment, a method was needed to convince pet dogs that their non-epileptic owners were about to experience an epileptic seizure. One possibility was to place gauze squares containing seizure-associated sweat samples into the pockets or socks of owners, but this proposal was rejected because of concerns about residual odours remaining in clothing [38]. Therefore, to prevent contamination issues, two specially designed pieces of apparatus called remote odour delivery mechanisms, (RODM, Figure 1), were used to deliver target odours from a separate laboratory to the dog owner’s location. Contamination issues would no longer be a problem because only target odours would be delivered from samples stored separately, which, after each trial, could easily be vented externally, leaving no residual trace.

Each RODM consisted of a pump attached to a scent container with an outlet pipe which delivered odours a distance of 6 metres from a different room with closed door. Validation of the RODMs was completed in a separate earlier investigation using operational police drugs dogs and game flushing field trial champion dogs [8]. The results validated that all the dogs recognized their specific target substances and responded to them as they had been trained with no lack of performance [8]. To further reduce risk of contamination, one RODM was used to deliver experimental odours and the other for controls with no seizure-associated odour. Thus, the RODM offered a simple but effective solution to the issues of cross contamination and in future may prove useful in preventing the risks associated with experimental procedures involving volatile or toxic materials.

### 2.5. Data Collection

Dog characteristics recorded were breed, sex, age-group and years owned. Indicators of seizure odour response by the 19 untrained dogs were those which were most frequently reported in other studies of this topic; intense staring at the owner, close-proximity to the owner, and pawing or nudging the owner [1,2,3,4,5,8,48,52,53,54,55,56,57,58].

Here, close proximity to the owner was measured as being within one metre as delineated by visual reference to marks which were already present on the floor of the laboratory, one to the left and one to the right of the owner. In studies of seizure alerting behaviours in dogs, owners were almost unanimous in believing their dogs’ pre-seizure behavioural changes were a warning mechanism [1,2,3,5,8]. Thus, while it is acknowledged that other potential behavioural responses such as avoidance, stress or aggression might also have been included in this study, they were excluded because they accounted for less than 2% of reported behaviours in several surveys of untrained canine seizure alerting activity.

All trials of the dogs were conducted ‘blind’, thus, neither the principal researcher nor the dog owners, had any knowledge of the sequences of odour presentation. The dogs’ responses were recorded in seconds and were measured over five trials, each of which lasted three minutes with breaks of two minutes between trials, to minimize the fatigue factor on the dogs [25]. Thus, including the 3 min habituation time at the beginning of a test series, each test session took approximately 28–30 min to complete. Recordings were by HP laptop installed video camera with a backup provided by an iPhone tablet camera. Analysis of the video footage was made without knowledge of the sequence of sample odours used to prevent unconscious confirmation bias [59]. Because only three behaviours were being monitored in this study, automated video tracking software was considered unnecessary.

### 2.6. Experimental Procedure

To minimize risk of pseudoreplication, prior to testing, one epilepsy volunteer from the three available was chosen as the initial source of samples for the day’s tests. This was done by means of a simple toss of the coin (following the procedure explained in ‘Test’) by the research assistant, who did not share the outcome with any other research member. The samples used included two controls and three seizure phase odours comprising pre-seizure, seizure, and post-seizure, (thus, 9 seizure samples in total). The samples from each of the three volunteers were assigned numbers:–Volunteer A: pre-seizure, 1, seizure, 2, post-seizure, 5–Volunteer B: pre-seizure, 3, seizure, 4, post-seizure, 6–Volunteer C: pre-seizure, 10, seizure, 11, post-seizure, 12–Control 1: 7–Control 2: 8

An online randomizer was used to produce several groups of five odour presentations for each of the epilepsy volunteers, consisting of their three seizure-associated sweat samples and two control sweat samples. The odour samples were re-used among the 19 dogs in a randomized fashion and were re-capped after each three-minute exposure. Thus, all the dogs were exposed to 9 seizure-associated samples but not necessarily all from the same person.

Individual dog owners and their dogs were assigned specific times to attend the testing laboratory at Queen’s University Belfast, and on their arrival and before any testing began, each dog was allowed to familiarize itself with the test area. Each dog then underwent a series of randomly delivered odour presentations via the RODM (see above) to an area beneath the dog’s owner who was seated in the middle of the test area and was asked not to engage with their dog. It was recognized that this lack of response was not natural and carried some risk of negatively impacting the dogs’ normal behaviour, but, had they been allowed to engage with their dogs as normal, they may have inadvertently influenced their dogs’ responses. The RODMs individually delivered the experimental and control odours following the sequences chosen by the research assistant.

The target behaviours being measured were *stands or sits and stares at their owner*—time eye contact, (TEC), which in one published work accounted for 70.8% of all attention seeking behaviours reported by over 130 surveyed dog owners [4]. The second attention seeking behaviour used in this study was time near owner (TNO), which [4], report accounted for (64.6%) of ASBs. The third response was time pressing close (TPC), which translates to nudging or pawing, accounting for (60.0%) of the behaviours identified in their survey [4].

Pre-test: On arrival each dog was individually brought to the test room which measured 10.5 m × 8 m × 4 m (Figure 1) and was given 3 min to habituate to the surroundings prior to the trial beginning.

Test: At the start of each day’s trials, the research assistant tossed a coin to determine which of the three volunteers’ samples would begin that day’s test, (A = 1, 2, 5, 7, 8/B = 3, 4, 6, 7, 8/C = 10, 11, 12, 7, 8). Thus, the coin was tossed three times using H to denote the winning value heads and T to denote the losing value Tails. The choices were A, B, C. If, after three tosses the outcome was HTT, then A wins. If the outcome was THT, then B wins and if the outcome was TTH, then C wins. In the event of a TTT or HHH outcome, the coin was tossed again until an uneven result emerged.

Six sets of randomized combinations of 5 sample sequences, were created for each of the three volunteers, for example, the sequences of sample odours for Volunteer A were, 1, 2, 5, 7, 8; 8, 2, 1, 5, 7; 7, 8, 2, 5, 1; 5, 2, 8, 1, 7; 2, 5, 8, 7, 1; 2, 1, 7, 5, 8. Once the initial toss of the coin had determined which of the three epilepsy volunteers would be used to begin that day’s trials, subsequent odour deliveries from volunteer one, two or three, were chosen at random by the research assistant. Between each odour presentation, the dogs were removed from the test area to rest for two minutes, while the room and RODMs were being ventilated as per the times shown below.

The two RODM scent chambers and pumps, one to deliver experimental odour and the other to deliver control odour, each at separate times, were kept in a separate laboratory to prevent risk of contamination. In addition, fresh sterile gloves were worn each time samples were handled. Each RODM (Figure 2) consisted of a new aquarium pump (5 W, 240/50 Hz Air pump 200 delivering 200 L/h) and connected to a re-sealable 3.6 L plastic watertight storage keg (UN approved, meaning, they are suitable for a range of applications such as transportation of chemicals to food or water storage). The storage keg had a resealable open top wide mouth and had inlet and outlet valves fitted at opposite sides. Pumps and kegs were connected to each other by 4 mm (internal diameter) plastic aquarium hose. The outlet pipes were 15 m aquarium tubing (4 mm) each with a nonreturn valve at the end. The outlet tubes were placed one under each thigh of the participant with the tube ends just showing at the inside of each leg. It was expected that this arrangement would increase the likelihood of the emerging scent samples remaining close to the participant’s body.

During exposure to each odour, the times spent by the dogs on each of the 3 behavioural responses, detailed above, were recorded. The lengths of time for which the pumps ran while delivering scent samples and during the system flush sequences were calculated thus:–Pump delivers 200 Lt in 60 min = 18 s/1 L–Airtight keg volume = 3.6 Lt–Time to clear 3.6 Lt = 3.6 × 18 s = 64.8 s = 1 min approx.–Time to run sample scent before introducing dog = 1 min.–Time to run sample scent, dog in room = 3 min–Flush time after trial (directed outside window dog out of room) = 1 min.–Total time for each scent sample = 5 min–Total number of samples per dog = 5 samples–Total time needed for each dog = 5 × 5 = 25 min–+3 min initial habituation time = 28 min/dog

### 2.7. Statistical Analysis

All data analysis was performed using the statistical package IBM Corp. Released 2016. IBM Staistics for Windows, Version 24.0. Armonk, New York, IBM Corp.

SPSS, (Version 24) IBM, Corporation, Armonk, NY, USA). Data from the two control odours, (C1, C2) were combined and averaged to give a mean for each of the three selected behavioural responses, time near owner combined control, time in eye contact combined control, and time pressing close combined control. The dogs’ three behaviours, time near owner (TNO), time in eye contact with owner (TEC), and time pressing close to owner (TPC) were then measured across all three seizure-related odours and comparisons drawn with their responses to the combined controls.

Visual inspection of histograms and use of Kolmogorov–Smirnov tests revealed the behavioural data were not normally distributed, thus indicating the need for nonparametric statistics. Friedman tests were used to examine whether each of the three behavioural responses (TNO, TEC, TPC) differed across the four treatment conditions, (pre-seizure, seizure, post-seizure and control). Wilcoxon signed rank tests were then used to make pairwise comparisons between each of the pre-seizure, seizure, post-seizure, with the mean control where significant differences were found.

## 3. Results

### 3.1. Demographic Information

As can be seen in Table 1, nineteen owners (16 female, 3 male), from a wide age range volunteered for this study, most of whom were female. The majority of the participating dogs were female, of pedigree status, and under 8 years of age. Most owners stated their dogs had been with them for 12 months or less and their main reason for acquiring a dog had been companionship. None of the dogs had witnessed an epileptic seizure and none of the dog owners had epilepsy.

### 3.2. Behavioural Responses to Seizure-Related and Control Odours

TNO—Time near owner differed across the four treatments, pre-seizure, seizure, post-seizure and combined control, (Friedman test, X^2^ (3) = 10.5, *p* = 0.015). The median levels for all four odour responses measured in seconds were respectively, 37.0, (IQR = 43), 47.0, (IQR = 71), 53.0, (IQR = 76), and 34.5, (IQR = 27) (Figure 3). More specifically, Wilcoxon paired comparisons of behavioural changes to the seizure-related odours with the combined control odour revealed: TNO pre-seizure, Z = −1.53, *p* = 0.13; TNO seizure, Z = −3.1, *p* = 0.002, TNO post-seizure, Z = −2.8, *p* = 0.005. Thus, dogs spent more time near their owner during the delivery of the seizure and post-seizure odours compared to the control condition, with no difference for the pre-seizure condition.

TEC—Time in eye contact: the dogs engaged in significantly more eye contact with their owners across the three seizure odours than they did with the combined control odours, (Friedman test, X^2^ (3) = 11.8, *p* = 0.008); median levels for TEC pre-seizure, seizure, post-seizure and TEC combined control were, respectively, 3.0 (IQR = 7.0), 7 (IQR = 9), 8.0 (IQR =11) and 2.5 (IQR = 5) (Figure 4).

Wilcoxon signed rank tests comparing response to seizure-related odours and the combined control revealed, TEC pre-seizure: Z = −2.3, *p* = 0.028; TEC seizure, Z = −2.2, *p* = 0.022 TEC post-seizure Z = −2.9, *p* = 0.004. Thus, dogs spent more time engaging in eye contact with their owners during the delivery of each of the three seizure related odours compared to the controls.

TPC—Time pressing close: was found to differ across the four treatments (Friedman test, X^2^ (3) = 8.4, *p* = 0.038); median levels for TPC pre-seizure, seizure, post-seizure and combined control were 4.0 (IQR = 7), 6.0 (IQR = 18); 1.0 (IQR = 8) and 3.0 (IQR = 5) (Figure 5). Wilcoxon signed rank tests comparing the responses of the dogs to the seizure-related odours and the combined control revealed, TPC pre-seizure, Z = −1.17, *p* = 0.24; TPC seizure, Z = −2.5, *p* = 0.013, TPC post-seizure, Z = −0.44, *p* = 0.66. Thus, dogs spent more time pressing close to their owner during the delivery of the seizure odour compared to control, with no difference for the other seizure related odours.

## 4. Discussion

This study explored the propensity of pet dogs to anticipate and respond to human epileptic seizure onset apparently emanating from their owners. Each dog was exposed to odours from three phases of seizure and their reactions compared with exposure to control odours. It was hypothesized that when seizure-odours were apparently coming from their owners, the 19 untrained pet dogs would demonstrate significant behavioural changes concomitant with attention-seeking activities. In line with this prediction, all nineteen pet dogs engaged in a significant increase in attention-seeking behaviours on detecting odours from seizure associated sweat samples compared with control odours. These findings have since been strengthened by a study which reports that seizures are associated with specific VOCs, which can be detected by trained seizure alert dogs more than an hour prior to a seizure’s manifestation [29]. In a separate investigation, further support has emerged from reports that a distinct VOC seizure-related profile has been detected by ion flow tube spectrometry (SIFT-MS) up to four hours in advance of a seizure [28]. Interestingly, [60] contend that a seizure-specific odour has been identified which is ‘predominantly of menthone’ [60] (p. 8), and this is also emitted by non-people-with-epilepsy who are experiencing a fearful situation and is believed to be an alarm pheromone. Taken together therefore, the results of this study and those of the published articles referred to, provide compelling support for the hypothesis that untrained dogs respond to epileptic seizures. This is an important finding because it offers the means to achieving a simple and reliable protocol for training dogs to warn people of an impending epileptic event, thus meeting a profound and long held wish for some form of pre-seizure warning device, held by those who live with difficult to control seizures [15,61]. It also holds out the hope, not only to improve patient safety and well-being but also of promoting ‘*therapies aimed at rapidly treating seizures (and) be able to abort seizures through targeted therapies*’ [62].

The behavioural responses of the dogs varied during the different seizure-phase sample presentations, but at this stage it is unclear whether the dissimilarities were an outcome of the phase of seizure, the sample taking procedure used, or the nature of the behaviour itself. For example, the ‘time spent near the owner’ (TNO) reaction differed significantly from controls during the seizure and post-seizure odours, but not during the pre-seizure odour presentation. It is possible this may have been the result of a sampling error by the volunteer or because there was an insufficient concentration of odour to meet the threshold for that response. For instance, the pre-seizure sweat sample capture depended upon recognition by the owner or family member, that their dog was displaying his/her innate ‘warning’ of an impending seizure event. Thus, a misinterpretation of the dog’s actions prior to prodrome, could have resulted in a less than optimum sample. A further anomaly was the unexpected similarity of the dogs’ reactions to the seizure and post-seizure odours, despite the post-seizure sample having been taken six hours after seizure. This outcome appears to contradict the findings of two recent studies [28,29]. The former report 3–4 dogs distinguished between seizure and inter-seizure samples (3 h post-seizure), with a probability of 93.7% [29]. Equally, [28] delineate inter-seizure samples, analyzed by SIFT-MS chemometric analysis taken 6 h post seizure, as having a significantly different VOC profile to that of seizure odour. This inconsistency may be understood, however, by considering the potential impact of environmental differences in sampling. For example, the volunteers in both the [29] study and that of [28] were all patients receiving treatment for epilepsy in medical centres with excellent ventilation and well-established clinical attention to cleanliness and hygiene. In the period following a seizure, therefore, it is conceivable that seizure scent could have quickly dissipated or that the patients may have been given a change of clothes or been moved to a different ward. In any of those circumstances, residual seizure odour would have been unlikely to linger, thus implying that the post-seizure samples used by those two studies, [29], study and [28], more accurately represented that phase of seizure. On the contrary, the sampling procedures used by the three epilepsy volunteers in this current study were not clinically derived, having been secured in the volunteers’ own homes before, during and after a seizure event. It is therefore conceivable that the 6-h post seizure samples had been contaminated by residual seizure odour which might explain why the 19 dogs in this current study gave positive responses to them instead of ignoring them as had been expected. Future research in this area, where clinical sampling of seizure associated samples may not be possible, would be advised to encourage volunteers to change their clothing between seizure stages, open windows to ventilate the room or even move to a different room, and, to allow a longer post-ictal sampling period, perhaps in the order of 12 h.

On the other hand, the behavioural response described as time of eye contact (TEC) by the dogs was found to be the most prolific of all three behaviours, having significantly increased across the three seizure phases, pre-seizure, seizure, and post-seizure, when compared to controls. This outcome is consistent with the findings of [4], who report that staring is the most common (70.8%) attention seeking mechanism engaged in by dogs. It also accords with [52], who found that gazing by dogs is often a request for help in the context of unsolvable tasks and reflects the findings of [58,63], who contend that gazing may have been a successful coping strategy in the past. Perhaps in the context of seizure odours, regardless of seizure-phase, the TEC response becomes the dogs’ preferred strategy in the face of what might be seen as an insurmountable problem [63].

The third behavioural response considered in this study was time pressing close (TPC), comprising of nudging or pawing their owners. The data suggest that this was significantly increased during detection of the seizure-phase odour compared to control, but no difference was found between controls and pre-seizure or post-seizure phase odours. These disparities are difficult to explain but might be understood in the context of TPC being the least frequently reported of the three attention-seeking behaviours. It is also the behaviour which is reportedly discouraged as most annoying by many dog owners [4]. That said, there is currently no way of knowing whether the potency of odour, regardless of seizure-phase, might fall short of a necessary threshold for evoking any particular behavioural expression, and would therefore indicate the need for further exploration.

In this investigation, seizure associated odours were presented to pet dogs using a novel method which was designed to deceive the dogs into believing the seizure-related odours were emanating from their owners. Thus, the current study demonstrates support for the hypothesis that a seizure-related olfactory trigger mechanism evokes spontaneous seizure alerting behaviour in pet dogs. This does not, however, exclude the possibility that other trigger stimuli may also exist, and, whilst beyond the scope of this study, it is acknowledged that other researchers have hypothesized the existence of a sensitivity in dogs to electromagnetic changes [21,64,65,66]. That being the case, it is conceivable that dogs may also be responding to electromagnetic signals which are associated with pre-seizure physiological changes [67,68,69,70,71,72] and invites further investigation. Thus, it is also conceivable that dogs which anticipate seizure onset may be responding not only to olfactory stimulation but also to minute variations in the body’s electromagnetic field which accompany the onset of epileptic seizure.

## 5. Limitations

It is acknowledged that a larger sample of seizure volunteers would have strengthened our findings, but despite a widespread appeal, we were unable to obtain any more than the three. They were, however, all from different countries, thereby reducing potential cultural bias.

All three seizure volunteers were female which was governed by participant response and therefore possible gender bias cannot be dismissed.

Seizure-odour degradation was a potential limitation which was addressed by recapping scent containers between each individual test and by restricting testing 3–4 dogs each day.

## 6. Conclusions

This study set out to scientifically examine anecdotal reports of seizure anticipation behaviours by untrained pet dogs and hypothesised that the trigger mechanism for these reported activities might be some form of odour that is unique to seizures. Consistent with this hypothesis, the results of this investigation provide compelling support for the contention that pet dogs anticipate epileptic seizures, consistent with an innate response to the impact of an olfactory trigger mechanism. These findings reflect those of more recent research into this subject. We have also provided compelling evidence for the contention that this olfactory biomarker(s) is directly associated with epileptic seizures, across their three phases, pre-seizure, seizure, and post-seizure. Although the nature of the biomarker remains unexplained, the findings have significant implications for developing a programme of targeted training for seizure prediction dogs with the possibility of a reduction in accidents and injury caused by unexpected seizure occurrences. The insights gained from this study may also be of assistance in improving the sense of self-worth of people living with a challenging epileptic condition while at the same time improving their quality of life and their sense of independence.

## Figures and Tables

**Figure 1 animals-11-02267-f001:**
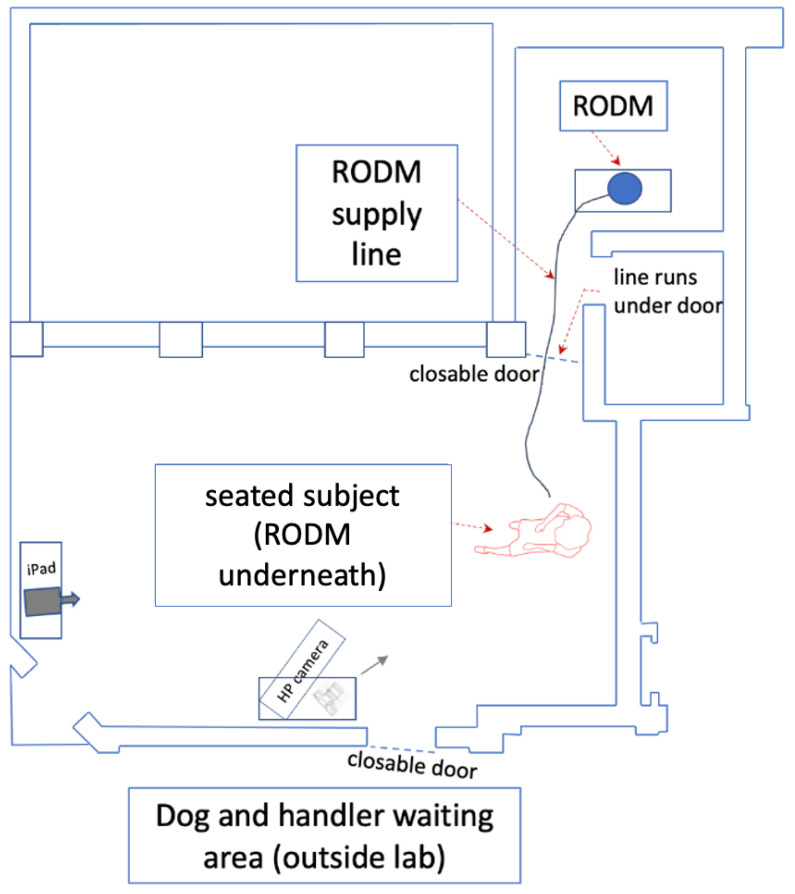
An overview of the test room used and arrangement for the RODM test. The test room measured 10.5 m × 8 m × 4 m.

**Figure 2 animals-11-02267-f002:**
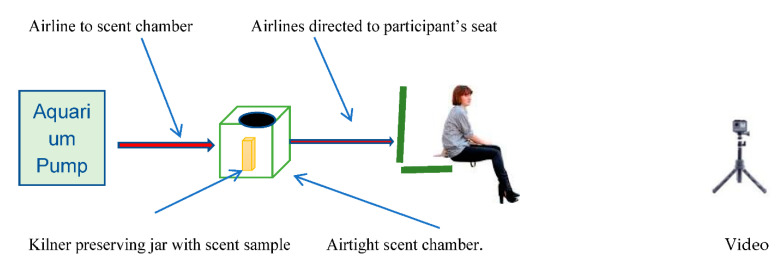
A schematic drawing showing how the remote odour delivery mechanism (RODM) was used.

**Figure 3 animals-11-02267-f003:**
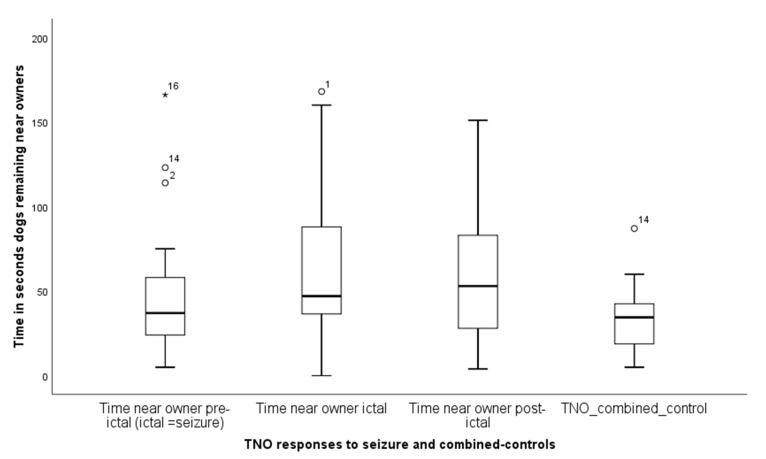
A box and whisker plot to show the times when the dogs remained in close proximity to their owners (one meter), measured across the four odour deliveries of seizure and combined control. The asterisks and circles represent outliers, that fall more than 1.5 times the interquartile range above the third quartile or below the first quartile.

**Figure 4 animals-11-02267-f004:**
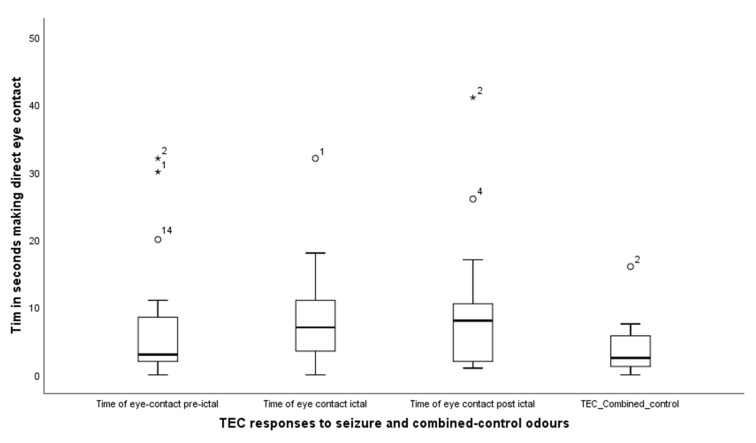
Showing the times when the dogs engaged their owners in a direct staring response across the 4 conditions, of seizure and combined control odours. The asterisks and circles represent outliers, that fall more than 1.5 times the interquartile range above the third quartile or below the first quartile.

**Figure 5 animals-11-02267-f005:**
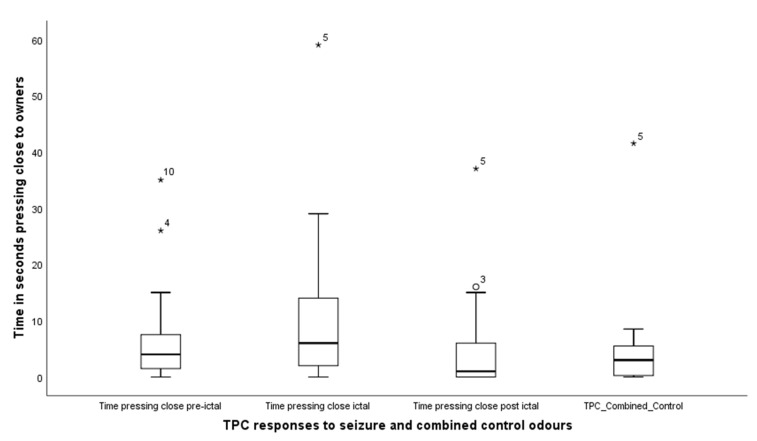
Box and whisker plot indicating the times spent by the dogs engaged in pressing close (TPC), to their owners by pawing or jumping up on them. The asterisks and circles represent outliers, that fall more than 1.5 times the interquartile range above the third quartile or below the first quartile.

**Table 1 animals-11-02267-t001:** Owner and dog demographics.

Demographic Factor Percentage	Number
Owner sex	
Male	3
15.80%	
Female	16
84.20%	
Owner age years	
18–25	1
5.30%	
26–35	7
36.80%	
36–45	3
15.80%	
46–55	1
5.30%	
56–65	6
31.60%	
66+	1
5.30%	
Dog sex	
Male	8
42.10%	
Female	11
57.90%	
Dog age months	
6–35	7
36.80%	
36–65	5
26.30%	
66–95	3
15.80%	
96–125	1
5.30%	
126–155	2
10.50%	
156–185	1
5.30%	
Dog breed	
Pedigree	13
68.40%	
Mixed	6
31.60%	
Length of ownership	
<12 months	13
76.50%	
1–5 years	6
23.50%	

## Data Availability

Data is available on request from the corresponding author. The data are not publicly available due to sensitivity over security issues concerning the involvement by Police Service NI dog handlers as part of the validation test.

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
