# Peer review of "The Untrained Response of Pet Dogs to Human Epileptic Seizures"

_animals, 2021, doi:10.3390/ani11082267_

Round 1

Reviewer 1 Report

Abstract: L10-12: "We hypothesised that seizures are associated with specific volatile organic compounds resulting in detectable odours which are the biomarker that triggers these reported behavioural changes in the dog" : as I understand it (not a native speaker though), the sentence is not accurate: previous work already suggest that VOC exists, the interesting hypothesis here, in my opinion, is that it could triggers the behavioural changes". I suggest to rephrase it in this sense. Also, it will fit better with the last sentence of the abstract (L22-23)

Introduction : L75 no training, and no experience? Making the distinction between an explicit training per se (i.e. the dog did not undergo the X months of training to be a certified seizure detection dog) and experience (i.e. the pet dog live with the epileptic person and, though not trained in anything related, has already been exposed to at least one humain seizure). As you speak here about totally naive dogs (that not even have seen one seizure in their lifes), I think it is important to stress it so the reader will not misunderstand.

L122 Result is difficult to generalize as only three persons whom their dogs alerted were sampled. Maybe VOCs are specific to persons/seizures/other factors and can not be generalized across "all people with epilepsy"

L148 : With gloves? If so, it would be a great additional information, if not, a limit to mention.

Sample collection & storage : how many samples were taken in total (and/or per person sampled)? Were some (or all) of them reused between the 19 dogs? It is suggested by the mention of recapping between presentation (L157). If it has been reused, how did you ensure that there still was an odour actually send to the dog after the 2nd, 3rd or more presentation?

Study participant: L165-167 : I am not sure this sentence is relevant in an article: this study had no direct benefice for the persons or dogs involved, and is not dedicated to demonstrate the existence of an odour but rather to investigate the behavioural changes of the dogs, that could be linked to odours exposure.

Data collection: L190 : "Were measured over five trials, each trial lasting 190 three minutes with breaks of two minutes between trials" Thus, including the 3 minutes habituation time at 192 the beginning of a test series, each trial took approximately 28 – 30 minutes to complete. I think there could be a différent word for a trial of 3min and a « session » ( ?) of 28-30min. It could otherwise be confusing for readers.

Figure 1: What are the green square and red lines on the right of the figure ?

Figure 2: In the version I have, lines from arrows are over the text.

Results: According to your results, and the importance of IQR, I would suggest to check inter-individual variability, if there are, I would be interesting to present it here as the purpose of the study is to explore untrained pet dogs reaction regarding seizure odours.

Conclusion: The conclusion is well written and sound, supported by the results.

Author Response

Reply to the reviewer's comments is in the attached file.

Reviewer 2 Report

Review Report:

Authors seek to confirm anecdotal reports that some untrained pet dogs are able to alert their owners to pending seizures by using a device developed by the authors that creates the illusion (to the owners' dogs) that a test scent or a control scent is being emanated by the dogs' owner. The test scents include pre-ictal; ictal; and 6-hour, post-ictal sweat (from 1 of 3 persons with epilepsy) versus a control scent from a person without epilepsy. Results show increased alerting behaviors to the seizure scents by the owners' dogs, despite the owners not having epilepsy, and presumably the dogs being naïve to seizure scent.

Comments:

This study is fascinating and a much-needed bridge between the desire for certified and trained epilepsy alert dogs and the supply (and affordability) of trained epilepsy alert dogs. As 1% of the world's population is thought to have epilepsy, and 1/3 of those are refractory to medications, the demand for alert dogs is enormous. While not discussed, the implied other half of the equation being explored in this study is the owners' ability to recognize the communication being offered by the dogs.

1) the topic is well reviewed and the literature cited is thorough (items #9 and #11 are duplicated in the version I have, but as numbers aren't utilized in the manuscript the reference count shouldn't disturb the manuscript).

2) I was a little confused about the testing procedure and who was being tested by reading the abstract alone; it became more clear as I continued to read that the dog in the non-epilepsy-owner:dog pair was being presented seizure scent from an unrelated epilepsy patient in order to fool the dog with the "novel" odor. I am not familiar with the terminology "bespoke apparatus", but if this is a term of art, then that explains my initial confusion. I would say for a more general/lay audience this term may need to be clarified.

3) It would be interesting, if authors agree, to include Maa et. al's follow up paper in the DISCUSSION. https://doi.org/10.1016/j.yebeh.2021.108078 "Epilepsy and the smell of fear" where the identity of the "unique" seizure scent is revealed and origin is explored. More specifically, a possible future work looking at using fear-scented sweat in the RODM apparatus to compare dog behaviors in both conditions. The hypothesized similarity of responses may be explained by alarm pheromone communication (alerting) and may explain potential differences in pre-ictal/ictal/and post-ictal sweat. As dogs and humans have evolved together (domestication to guard humans while sleeping) it makes sense that dogs would be keyed into human threats.

Author Response

(The authors gave the same response as above.)

Reviewer 3 Report

Searching for reliable warning methods of upcoming seizure onset, is without doubt a desitable goal. Such methods would improve the quality of live and safety of people with seizure, as well as the quality of life of families of the patient. First reports in peer-reviewed scientific journals on the phenomenon of seizure-reacting or seizure-alert dogs, were published more than 20 years ago, however, it is still not clear what is the mechanisms and biological role of this ability of canines. Some specific canine behaviors were identified in pet dogs which learned spontaneously to react and/or to predict seizure in their owners or in family members. It was also demonstrated that dogs can be trained for this purpose. Whereas the early published papers, based mainly on questionnaires, hypothesized that dogs react to subtle changes in the behavior of humans that characteristically precede a clinical seizure or react to patient’s symptoms that are overt during the seizure onset, the other consecutive studies hypothesized that the dogs may react to putative human odor emitted before and during the seizure onset, or react both to the odor and behavior of the patient. The reviewed study uses experimental approach to investigate if untrained pet dogs are able to distinguish and to react to human odor emitted before, during and after seizure onset. The experimental setup is very original, consisting in presenting to the dogs  the sweat odor of patients with seizure collected shortly before, during and after seizure onset and delivered to the dogs using a special device called Remote Odour Delivery Mechanisms (RODM) imitating that the odor of seizure patient would be emitted by the dog owner who has no seizure. The authors found significant differences in duration of three most frequent behavioral reactions in seizure-reacting dogs reported in the literature, as assessed in the present experiments between trials using sweat odor of patients with seizure (pre-seizure, seizure and post-seizure odor samples) and samples from healthy controls. From this finding the authors conclude that a seizure-related olfactory trigger mechanism evokes spontaneous seizure alerting behaviour in untrained pet dogs which had never experienced seizure onsets in their owner who were obviously healthy. The adopted experimental setup and the material, however, seem not to exclude a bias, which should be critically adressed in the discussion section.

It could be a difference between dogs that are actually untrained, meaning that they never experienced a seizure onset in their owner or in family members and are exposed for the the hypothetical seizure odor for the first time, and on the other hand, dogs which are formally „untrained” but experienced seizure onsets several times in people with whom they are familiar or bonded to. A seizure onset with its all accompanying behaviours of the patients like sudden falling down, uncontrolled jerking movements, convulsions etc. could be annoying for dogs, causing that the dog demonstrate behaviours as described in the literature. If a seizure onset is really accompanied by a specific change in odor emitted by patient’s body, even prior to the onset, the dogs can be „self-conditioned” to the seizure, which is then interpreted by people as „alerting”. For dogs that never experienced seizure in humans, the putative seizure odor has no biological meaning or value, is neither associated with a reward or punisment, however, might be simply interesting due to its novelty.

The one bias may be due to the experimental setup where the odors of sweat samples collected from 3 humans with seizure are pumped via the RODM to the room where the healty owner of the dog sat.

Thus, the odor plume around the dog owner was a sort of mixture of the dog owner odor, which is well known to the dog, and the putative „seizure odor” plus individual odor of an alien human (patient or healthy control). This situation could be novel, surprising and confusing for the dog which may try to examine more exactly its owner and/or may try to seek for additional cues. Such situation may involve longer TEC, TNO and TPC.  Usually there is a clear effect of odor novelty as manifested by decreasing interest of the dog in novel odors presented for the first or the second time compared to  further consecutive exposures. If, for example due to the randomization of the presentation of different odors to the dog, a tendency appeared by chance, that „seizure” odors were presented on average firstly i.e. before the control odors, this might result in longer TEC, TNO and TPC for „seizure” odors. This bias might not necessarily appear, but it would be useful to analyse consecutive trials starting with the first one.

The second bias may be related to a relative small number of odor donors investigated (only 3 patients with seizure and 2 healthy control persons).  In experiments on canine olfaction there is sometimes an issue of differentiated „attractiveness” of odors of individual humans to dogs, which translates in longer or shorter examining of „attractive” odors. If, for example by chance, the 3 patients with seizure had more „attractive” individual odors than controls, this may also have impact on differences in TEC, TNO and TPC between seizure patients and controls. This issue should be also addressed in the discussion as a limitation of the study

As for the seizure-alerting dogs, the real question is, if they are able to distinguish and to react to pre-seizure and seizure odors and ignore/not react to the body odor of the patients in periods when no seizure takes place. The authors did not collect odor samples in seizure-free periods, so this question cannot be answered using the present experimental setup. It could be that patients with seizure have generally different body odor from healthy subjects independently on the disease phase (pre-seizure, seizure, post-seizure). Comparing control odor samples (healthy) with pre-, seizure and post-seizure samples does not answer this question.     

 Minor remarks.

Some quantitative results should be given in the Abstract.

 L.80-100 - The text in lines 80-100 belongs rather to the M&M section than to the Introduction

  1. 124 - was the ability of the dogs owned by patients with seizure assessed by any quantitative data or just reported by the dog owners without any data? Were any false alerts of the dogs reported ?

L.129-130 - why odor samples from only 3 females with seizure were collected ? , were males with seizure not available for this study ?

L.138 - were  the two donors of control samples females or males ?

L.138-140 – why were as potential confounding factors only having/not having a pet dog and bathing/showering habits considered ? – what about other potential confounding factors as e.g. diet before sampling, medications, other diseases ?

L.144-146 - how were the sweat samples collected ? – from armpit or from palms ? how long was the gauze pad held during collellection ?

L.148-149 - „…pre-seizure samples were taken when untrained pet dog characteristically indicated that a seizure was imminent…” – were ALL pre-seizure samples taken in situations when the seizure actually followed the characteristic behavior of dogs in pre-seizure situations, meaning that there were no false alerts by dogs ?

Obviously no such question concerns the seizure and post-seizure samples because seizure could be ascertained by the family members or other people present during s seizure onset.

L.150 – who collected seizure samples while a seizure was occuring? - a family member, medical staff or somebody else. It is hardly imaginable that the patient herself would collect odor samples during seizure onset.

  1. 153 – how many samples were collected altogether from subjects with seizure and from heathy controls? Were the pre-seizure, seizure and post-seizure samples collected only once or on several occasions ?
  2. 209 - why the numbers 7, 8, and 9 were not assigned for the volunteer C ?
  3. 212 - why the control samples were not assigned any numbers?
  4. 233 - in my understanding tossing coin gives two alternatives – how it was transponed to three volunteers’ samples ?
  5. 234 The assignment of numbers is not clear: what do the numbers 7 and 8 denote ? , were the samples 7 and 8 the controls ?

L.237 – here are 6 randomized sets instead of 5 mentioned in line 235 ?

Figures – what do the asterics and small circles with figures denote ? The readers may not be familiar with whisker plots.

L. 351-352 - did ALL individual 19 dogs demonstrate increased time parameters associated with seizure odors, or there were only statistically significant differences as assessed by the Wilcoxon test ?

In summary, despite of some weaknesses as listed in my review, this paper has merits for better understanding of the phenomenon of canine reaction to seizure in humans. Therefore I recommend resubmission of this paper after major revision, in particular with addressing possible bias in the results.

Author Response

(The authors gave the same response as above.)

Round 2

Reviewer 3 Report

The revision has been substantially improved as compared to the first submission.  Many thanks for the rebuttals to my review and for explanations to my questions. I was nearly inclined to accept the revision without changes, however, I believe the article would benefit from a better structuring.

I mean first of all shifting the passage in lines 85-105 of the revision (lines 80-100 of the first submission), to the Material & Method section e.g. between lines 188-189 (before the paragraph Data collection), since this passage describes methodological aspects concerning the present study. The authors in their reply denoted this issue with question marks in red, which, in my understanding means that they either disagree with me, or have some doubts whether my suggestion is appropriate.

On the other hand, the passage marked in red, inserted in the revision M&M lines 159-165, fits better to the Introduction between lines 61 and 62.

Lines 29-31 In my understanding as a non native English speaker, strictly speaking, it was not a seizure onset that was simulated in this study, since there were no full seizure symptoms like falling down, convulsions etc, but the dogs were exposed to the odor that was deemed characteristic to seizure.

Figures – my question was: what do the asterics and small circles with figures denote ?  Authors’ Response was: „these represent outliers, that fall more than 1.5 times the interquartile range above the third quartile or below the first quartile”. – thank you for the clarification to me, but this clarification should be also inserted into the figure legends for a better clarity to the readers, who may not be familiar with whisker plot.

In conclusion, my remarks to the revision are of a minor nature and can be easily introduced to the final version of the manuscript, if the authors and the editor think they are necessary. Thus I believe my next review will not be necessary.

Author Response

Our sincere thanks to the minor revisions suggested by Reviewer 3. We have incorporated these into the article, which we believe, is much the better for them. The changes referred to are shown in red and are as follows:

Lines 32-35

Lines 164-167

Lines 170-191

Lines 342-345

We hope the above meet with your approval but, should any other issues emerge, please contact us. 
